

# Muscle synergies during the walk-run and run-walk transitions

Leonardo Lagos-Hausheer[1,2], Samuel Vergara[3], Victor Munoz-Martel[4], Germán Pequera[1,5], Renata L. Bona[1] and Carlo M. Biancardi[1]

[1] Biomechanics and Movement Analysis Research Laboratory, Department of Biological Sciences, CENUR Litoral Norte, Universidad de la República, Paysandú, Paysandú, Uruguay
[2] Department of Physiotherapy, Faculty of Medicine, Movement Physiology Laboratory, Universidad de Concepción, Concepción, Chile
[3] Electrical Engineering Department, Faculty of Engineering, Universidad Católica de la Santísima Concepción, Concepción, Concepción, Chile
[4] Department of Training and Movement Sciences, Humboldt Universität zu Berlin, Humboldt Universität Berlin, Berlin, Berlin, Germany
[5] Ingeniería Biológica, Universidad de la República, CENUR Litoral Norte, Uruguay

Corresponding author
Leonardo Lagos-Hausheer, lagoskinesiologo@gmail.com

## ABSTRACT

**Background**. Muscular synergies could represent the patterns of muscular activation used by the central nervous system (CNS) to simplify the production of movement. Studies in walking-running transitions described up to nine synergy modules, and an earlier activation of flexor and extension ankle muscular groups compared to running or walking. Our project aims to study the behaviour of muscle synergies in different stance and swing variations of walking-running (WRT) and running-walking (RWT) transitions.

**Methods**. Twenty-four trained men participated in this study. A variable speed protocol on a treadmill was developed to record the activity of 14 muscle during walking, running and relative transitions. The protocol was based on five ramps of 50 seconds each around ± 10 and 20% of the WRT speed. WRT and RWT were identified according to an abrupt change of the duty factor. Analysing surface electromyography using non-negative matrix factorization (NMF) we obtained synergy modules and temporal activation profiles. Alpha threshold for statistical tests set at 0.05.

**Results**. We described four different transition strides, two for increasing speed transitions, and two for decreasing speed transitions. Four to six synergy modules were found in each condition. According to the maximum cosine similarity results, the two identified WRT conditions shared five modules, while the two RWT conditions shared four modules. WRT and RWT overall shared 4.33 ± 0.58 modules. The activation profiles and centres of activation revealed differences among conditions.

**Discussion**. Transition occurred at step level, and transition strides were composed by walk-like and run-like steps. Compared with previous studies in running and walking, both transitions needed earlier activation of a comparable number of synergy modules. Synergies were affected by acceleration: during RWT the need to dissipate energy, to decrease the speed, was achieved by increasing the number of co-activating muscles. This was reflected in fewer synergy modules and different activation profiles compared to WRT. We believe that our results could be enforced in different applied fields, like clinical gait analysis, physiotherapy and rehabilitation, where plans including co-activation of specific muscular groups could be useful. Gait transitions are common in

different sports, and therefore also application in training and sport science would be possible.

## INTRODUCTION

It has been suggested that the central nervous system (CNS) coordinates muscle activity by means of modular structures called synergies, or modes (*Krishnamoorthy et al., 2003*), which represent co-activated clusters of motor units of different muscles.

Muscle synergies have been described using linear algorithms, like principal component analysis or non-negative matrix factorization, applied to surface electromyographic signals of a complex of several muscles (*Cappellini et al., 2006*; *Pequera, Ramírez Paulino & Biancardi, 2021*). They were investigated in different activities and exercises, and during symmetric and asymmetric locomotion tasks, including transitions (*d'Avella, Saltiel & Bizzi, 2003*; *Hagio, Fukuda & Kouzaki, 2015*; *Pequera, Ramírez Paulino & Biancardi, 2021*; *Santuz et al., 2022*; *Yokoyama et al., 2016*).

Four to five main synergies were identified for walking (W) (*Santuz et al., 2022*). Further studies indicated that these synergies were shared with running (R) (*Cappellini et al., 2006*; *Ivanenko, Poppele & Lacquaniti, 2004*; *Pequera, Ramírez Paulino & Biancardi, 2021*), and skipping (*Pequera, Ramírez Paulino & Biancardi, 2021*). The number of synergy modules may decrease in pathologic conditions (*Clark et al., 2010*; *Danner et al., 2015*; *Rodriguez et al., 2013*), or may increase during walk-run (WRT) and run-walk transitions (RWT) (*Hagio, Fukuda & Kouzaki, 2015*; *Kibushi et al., 2021*). However, during other kind of gait transitions involving W (stair or ramp ascent and descent) five synergy modules were described (*Liu & Gutierrez-Farewik, 2022*).

When new motor demands challenge the CNS, synergy modules can be adapted in a flexible way (*Kibushi et al., 2018*). The mechanics of walking and running are quite different: the inverted pendulum model was developed to describe W, while a spring-mass model better apply to R (*Saibene & Minetti, 2003*). During gait transitions adaptations would be necessary to adjust pace and speed (*Minetti, Ardigo & Saibene, 1994*). While increasing the W speed, the mechanical transition threshold toward R can be reached. WRT is associated to several changes, like angular values between thighs (*Minetti, Ardigo & Saibene, 1994*), or changes of the ankle articular range (*Farris & Sawicki, 2012*). The major change involves the transition from a pendular motion to a bouncing gait, and some authors predicted also metabolic cost effects (*Hreljac, 1993*). It has been suggested by Hagio and collaborators (*Hagio, Fukuda & Kouzaki, 2015*) that spontaneous WRT would be triggered by afferent information but, on the other hand, voluntary WRT would be only generated by descending neural input directly to actuators (muscle synergies). They determined synergies in WRT and RWT using bilateral electromyography and assuming the existence of a synergy coordination between the legs, therefore, the nine synergies described

corresponded to both extremities simultaneously. There are still open questions regarding muscle activations and synergies, especially when different modalities of transition would be involved. The increased number of synergies during WRT and RWT could be related to various factors, like the number of monitored muscles and their position (unilateral or bilateral), including the small sample size: five to eight subjects (*Hagio, Fukuda & Kouzaki, 2015*; *Kibushi et al., 2021*). Synergies information could be relevant to understand the behavior of the neuromuscular system. For instance, should a common motor pattern like that described by *Pequera, Ramírez Paulino & Biancardi (2021)* include gait transitions? Understanding the refinements of motor control and the behavior of muscle synergies is now, more than ever, crucial thanks to artificial intelligence. This is key to generating more accurate computational models that simulate motor control, predict movement patterns, and create more realistic representations of muscle coordination. The outcomes of such investigation can hopefully be applied to programming clinical and sports training sessions, which could include or not gait transitions based on the muscle groups involved in training or therapy. The main objective of our research was to determine which muscle synergies are evoked during different variations of WRT and RWT. Our hypothesis, based on the cited literature and on the mechanics of walking and running, is that to achieve this objective, we developed a variable speed protocol (VSP) on a treadmill, designed experiments to record the activity of 14 muscles through surface electromyography (EMG), and used scripts to extract muscle synergies using non-negative matrix factorization (NMF) (*Lee & Seung, 1999*; *Santuz et al., 2022*).

## MATERIALS & METHODS

The research was based on a cross-sectional study design.

### Participants

Twenty-four trained men signed an informed consent and participated in the research. The sample size was estimated using data from previous research (*Pequera, Ramírez Paulino & Biancardi, 2021*). The experiments were performed in accordance with the Declaration of Helsinki, and were approved by the conduct of this research was approved by the Ethics Committee and by the Council of the CENUR Litoral Norte of the University of the Republic (Exp . #311170-000921-19). Age, height, body mass, body mass index (*BMI*), average distance traveled per week were determined for each subject (Table 1).

### Data collection

To determine the transition speed (*Ts*) of each participant performed three trials on a treadmill (T2100; General Electric, Cincinnati, OH, USA). Starting from a comfortable speed of 0.8 m/s, the treadmill's velocity increased by 0.1 m/s every 15 s until the subject started running. The average transition speed of the three trials was used to create the variable speed protocol (VSP). The VSP was performed twice by each participant and consisted of five ramp cycles of 50 s. In each ramp cycle the treadmill's velocity was composed of ten stages of 5 s (*Ts*-20%, *Ts*-10%, *Ts*, *Ts* +10%, *Ts* +20% and reverse). This allowed the subjects to change from walking to running and from running to walking

**Table 1  Summary results of the analyzed sample.**

| Variable | Mean ± Std. dev |
|---|---|
| Age | 32. 53 ± 10.99 year |
| Weigth | 72.96 ± 9.51 kg |
| BMI | 23.08 ± 2. 91kg $m^2$ |
| Heigth | 1.75 ± 0.08 m |
| Kilometers per week | 47.11 ± 35.28 km |
| Years of training | 6.00 ± 4.51 year |
| Transition speed (Ts) | 1.98 ± 0.70 m $s^1$ |

several times during the protocol. A detailed description of the protocol was provided in a previous work (*Lagos-Hausheer, RL & Biancardi, 2023*).

Three-dimensional body motion was sampled at 100 Hz by an eight-cameras MOCAP system (Vicon; Oxford Metrics, Yarnton, UK), using a model of 18 reflective markers and 11 segments (*Pavei et al., 2017*).

The activity of 14 ipsilateral muscles was sampled at 2000 Hz with a 16 CH Delsys Trigno EMG systems (Delsys, Boston, MA, USA) synchronized with the MOCAP system. The electrodes were placed on the right side of the body following the SENIAM recommendations (*Stegeman & Hermens, 2007*). Muscles selection was based on literature (*Hagio, Fukuda & Kouzaki, 2015*; *Kibushi et al., 2018*; *Pequera, Ramírez Paulino & Biancardi, 2021*; *Santuz et al., 2017a*).

Ankle extensors: *Soleus* (SOL), *Gastrocnemius Medialis* (GASM), *Peroneus Lateralis* (PERL). Ankle flexors: *Tibialis Anterior* (TIBA). Knee extensors: *Vastus Lateralis* (VASL), *Vastus Medialis* (VASM), *Rectus Femoris* (RECT). Knee flexors: *Biceps Femoris* (long head) (BICF), *Semitendinosus* (SEMT). Hip extensors: *Gluteus Medius* (GLUM), *Gluteus Maximus* (GLUMAY), *Tensor Fascia Latae* (TFL). Spine erectors: *Longissimus* (LONG), *Iliocostalis* (ILIO).

## Kinematic analysis

Heel-strike (*HS*) and toe-off (*TO*) events were identified from the reconstructed vertical trajectory of the heel and 5th-methatarse markers (*Cappellini et al., 2006*; *Pavei et al., 2017*). The right *HS* was used to cut the locomotion cycles (strides), and the right *TO* allowed to separate the stance and the swing phases. The relative duration of the stance phase (DF) was used to identify the transition strides, as gait change is accompanied by an abrupt change of DF. Based on the DF of single steps, transition strides have further been split into four variants: (A) WRT strides composed by a walk-like right step (right stance phase) followed by a run-like left step (right swing phase); (B) WRT strides composed by both run-like steps; (C) RWT strides composed by both walk-like steps; (D) RWT strides composed by a right run-like step (right stance phase) followed by a walk-like left step (right swing phase) (Fig. 1).

## Electromyographic processing

Electromyographic data of the identified transition strides were processed. High-pass filter was used to remove unwanted low frequencies, followed by full-wave rectification

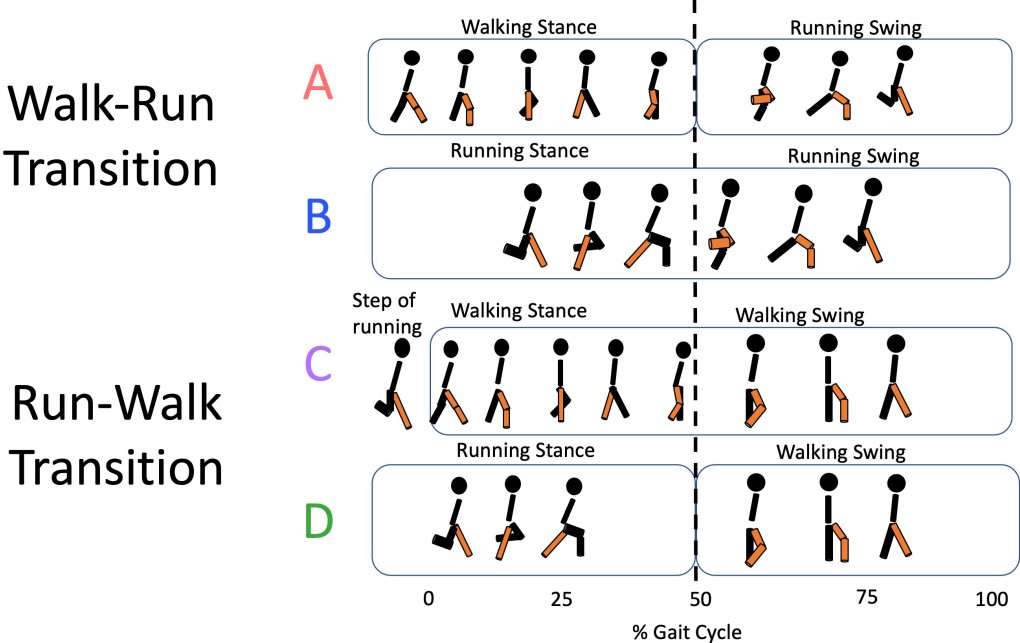

**Figure 1 Transition types.** For the walk-to-run transition, types A (walking and running steps) and B (running steps only) are identified; for the run-to-walk transition, types C (walking steps only) and D (running and walking steps) correspond.

and a low-pass filter using a fourth-order IIR Butterworth zero-phase filter with cut-off frequencies of 30 Hz (*Rabbi et al., 2020*) and 6 Hz (*Hagio, Fukuda & Kouzaki, 2015*) respectively. Amplitude normalization was performed by dividing each EMG signal by its maximum value recorded for each participant under all conditions. To standardize the time length of the walking cycles, they were normalized to 200 points, assigning 100 points for each phase, stance and swing (*Santuz et al., 2017a*).

Synergy analysis was carried out using a "R" script (R v4.2.3; *R Core Team, 2022*) (*Santuz et al., 2022*). We will summarize the main steps, as the whole process was described in literature (*Santuz et al., 2017a*; *Santuz et al., 2017b*). The script applies the non-negative matrix factorization (NMF) algorithm to a V ($m \times n$) matrix, where $m$ were the 14 muscles, and $n$ the number of data points. Resulting synergies were composed by the motor modules matrix (WM) and the motor primitive matrix (H). The former matrix ($m \times r$) stored the muscle ($m$) weightings of each synergy module ($r$) (*Santuz et al., 2017b*; *Santuz et al., 2017a*), while the latter ($r \times n$) stored the time-dependent coefficients of the factorization ($n$) of each synergy module ($r$) (*Dominici et al., 2011*; *Santuz et al., 2017a*; *Santuz et al., 2017b*). WM and H were used to obtain a reconstructed $m \times n$ matrix (VR), which approximates V (V ≈ VR). The goodness of fit between V and VR was measured by the coefficient of determination $R^2$. The minimum number of synergies required to represent the original signals was assessed after several repetitions by fitting the curve of $R^2$ values *versus* synergies with a linear regression model and calculating the mean squared error (*Cheung et al., 2005*; *Santuz et al., 2017a*).

## Centre of activation

The angle ($\theta t$), in polar coordinates, of the vector that points to the centre of mass of the circular distribution of a motor primitive was defined as centre of activation (CoA) (*Cappellini et al., 2006*). Thus, the CoA can take values that range from $\theta t = 0$ to $\theta t = 2\pi$, representing that interval the gait cycle. The CoA was computed for all the synergy modules in all conditions (WRT and RWT), by means of Eqs. (1), (2) and (3) (*Santuz et al., 2017a*):

$$A = \sum_{t=1}^{p}(cos\theta_t P_t) \tag{1}$$

$$B = \sum_{t=1}^{p}(sin\theta_t P_t) \tag{2}$$

$$CoA = arctan\left(\frac{B}{A}\right) \tag{3}$$

where $p$ is the number of points of each transition cycle and P is the amplitude of the activation vector.

## Statistics

Maximal cosine similarity (*cosim*) was used to measure the similarity between motor modules across different transition conditions. This measure represents the cosine of the angle between two weight vectors, and has been used for assessing similarity in symmetric and asymmetric gaits (*Hagio & Kouzaki, 2014*; *Pequera, Ramírez Paulino & Biancardi, 2021*). Given two vectors a and b, their *cosim* would be:

$$cosim(a,b) = cos\theta = \frac{\sum a_i b_i}{(\|a\|\|b\|)} \tag{4}$$

Values of *cosim* above 0.6 would indicate similarity between motor modules (*d'Avella, Saltiel & Bizzi, 2003*; *Pequera, Ramírez Paulino & Biancardi, 2021*). One sample $t$-test was applied on *cosim* values in each condition to check if they were significantly larger than the 0.6 (similarity threshold). Statistical significance was assessed at 0.05. Analyses were performed with the software PAST 4.11 (*Hammer & Harper, 2001*).

## RESULTS

### EMG envelopes

Spine erector muscles, LONG e ILIO, were mainly active during the stance phase and in the late swing phase, immediately before the HS, in all transition conditions, except for ILIO in RWT "D" variant (Figs. 1 and 2). In the hip extensors group, GLUM and GLUMAY, with the knee extensors VASL and VASM, were mainly active in the swing phase, before the HS, while TFL was also active during the stance phase, with differences among the four

transition conditions (Fig. 2). RECF was mainly active from the late stance phase to the early swing phase (Fig. 2). Activation of the knee flexors was variable across the different transition conditions (Fig. 2). TIBIA was active from the late stance phase to the swing phase, while the ankle flexor muscles were mainly active in the stance phase, with slight differences for SOL (Fig. 2).

## Muscle synergies

Figure 3 shows the number of synergies extracted during 20 repetitions of the script in each transition condition. Within 3.5 and 6 synergy modules were necessary to meet the reconstruction criteria required, the differences were significant (Kruskal-Wallis; $p = 0.005$).

Several synergy modules were composed by the same muscles across the different transition conditions (Figs. 4 and 5). The erector spinae muscles were included in a synergy module present in all conditions (Fig. 4, M2 in A, B and D; M1 in C). The same occurred for the knee flexors (Fig. 4, M5 in A; M4 in B; M3 in C and M6 in D). A module composed by the ankle extensor muscles was present in WRT and RWT (Fig. 4, M1 in A, B and D), but in the RWT "C" transition (both walking-like steps), ankle extensors were included in different synergy modules, associated with the knee extensors (Fig. 4C). Knee extensors were anyway the principal muscles in a synergy module of WRT (Fig. 4, M6 in A; M5 in B). Hip extensors were included in synergy modules associated with other muscles, and mainly in RWT (Fig. 4, M4 in C; M5 in D).

According to maximal cosine similarity results, conditions A and B of WRT shared five synergy modules, while conditions C and D of RWT shared four synergy modules. WRT and RWT overall shared $4.33 \pm 0.58$ synergy modules. Conditions with shared synergy modules were summarized in Fig. 5.

## Synergy temporal patterns

Figure 5A shows the distribution of the centres of activation (CoA) of the synergies within the transition cycle, in the four conditions (A to D).

During the first part of the stance phase, weight acceptance, the CoA of two synergies (C1 and C2) can be observed, within 2% and 27% of the normalized cycle in conditions A, B and D (Fig. 5A). During the second, propulsive part, of the stance phase was the C1 of condition C, while within 35% and 46% of the normalized cycle were found the synergies C3 of conditions A, B and D (Fig. 5A). In the early swing phase, within 59% and 72% of the normalized cycle, appeared the synergies C4 of A, B and D, and the synergy C2 of condition C (Fig. 5A). The last synergies, C5 (A, B, D), C6 (A, D) and C3-C4 (C), appeared in the late swing phase. Some of the CoA were located near the 90% of the normalized cycle (C6 of A and D, and C3-C4 of condition C); therefore, in a phase of preparation for the heel strike, remembering that the actual activation extends for a span of time before and after the location of the CoA (Fig. 5A).

In Fig. 5B the CoA have been grouped for muscular group. The CoA of the synergy modules that include a muscular group were displayed in the corresponding circular graph.

The CoA of the ankle extensors (PERL-GASM-SOL), included in the first synergy of three transition conditions (M1 in Fig. 4ABD), were located within the weight acceptance

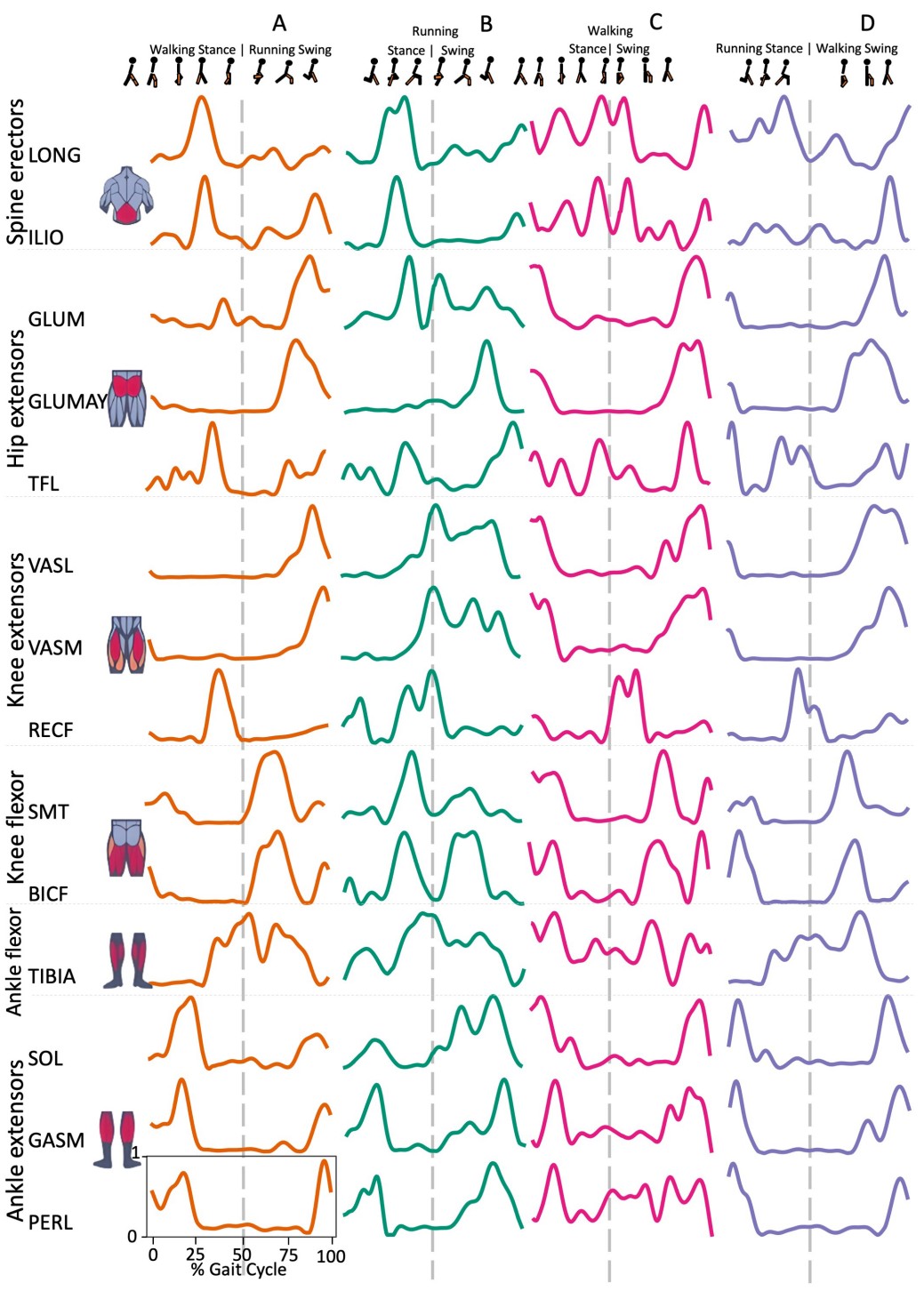

**Figure 2 Muscle activation.** Envelopes of the analysed muscles over the gait cycle, divided into stance and swing phases, for transition type (A to D). Envelope characteristics are described in methods. Muscles: LONG = *Longíssimus*, ILIO = *Iliocostalis*; GLUM = *Glúteus medius*; GLUMAY = *Glúteus maximus*; TFL = *Tensor fascia latae*; VASL = *Vastus lateralis*; VASM = *Vastus medialis*; RECF = *Rectus femoris*; SMT = *Semitendinosus*; BICF = *Bíceps femoris*; TIBIA = *Tibialis anterior;* SOL = *Soleus*; GASM = *Gastrocnemius medialis*; PERL = *Peroneus longus*. Muscle Group illustrations source credit: https://stock.adobe.com/es/images/muscles-illustration-icon-set-it-included-the-workout-human-body-parts-anatomy-and-more-icons/600376660?asset_id=600376660. Standard license, creator: antto.

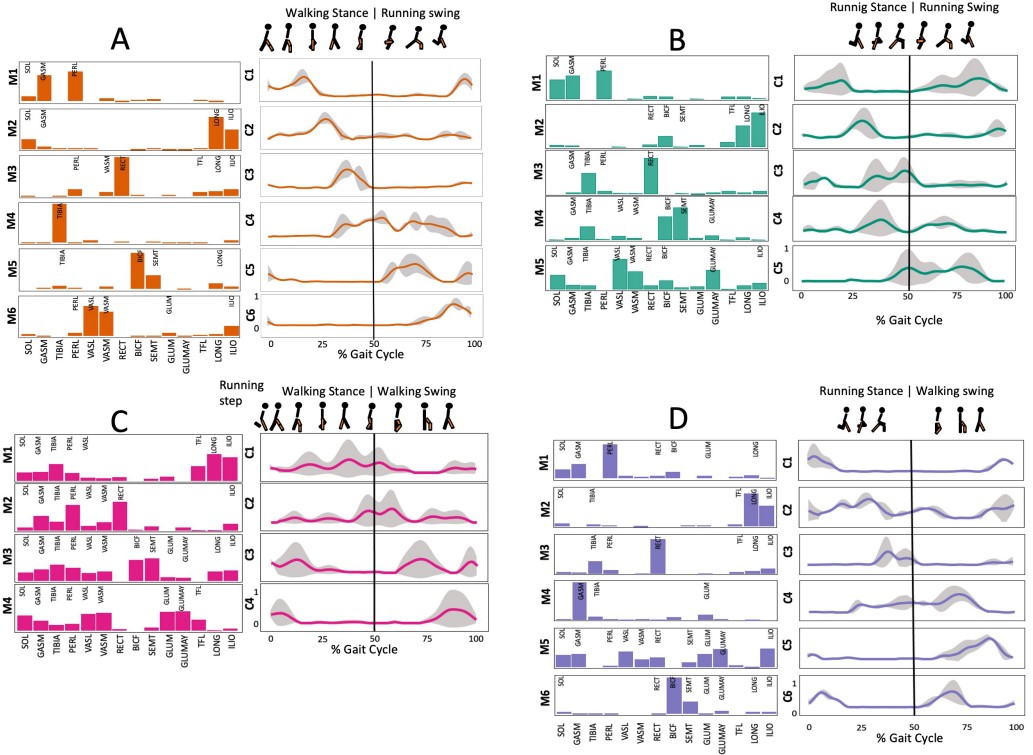

**Figure 3** **Muscle synergy modules and activation time profiles of WRT (A, B) and RWT (C, D).** Synergy modules (M1...Mx): bar heights represent the weight of each muscle within the module. Timings profiles (C1...Cx): the vertical line divides the gait cycle into stance and swing phases; solid lines represent the average activation profile of each module, while the shaded areas represent ±sd. Muscles: SOL = *Soleus*; GASM = *Gastrocnemius medialis*; TIBIA = *Tibialis anterior;* PERL = *Peroneus longus*; VASL = *Vastus lateralis*; VASM = *Vastus medialis*; RECF = *Rectus femoris*; BICF = *Bíceps femoris*; SMT = *Semitendinosus*; GLUM = Glúteus medius; GLUMAY = Glúteo máximo; TFL = *Tensor fascia latae*; LONG = *Longíssimus*, ILIO = *Iliocostalis*. Running and walking silhouettes source credit: Authors' original work.

phase, between 2% and 11% of the normalized cycle (Fig. 5B). Spine erector muscles were predominant in the second synergy of A, B and D (M2 in Fig. 4ABD), while in the transition condition C they were the principal muscles of the first synergy (M1 in Fig. 4C). These synergies were active around the midstance phase, between 16% and 33% of the normalized cycle (Fig. 5B). In the propulsion and pre-swing phases, between 35% and 52% of the normalized cycle, were activated the synergies where the RECT was the main muscle (M3 in Fig. 4ABD; M2 in Fig. 4C). Ankle flexors (TIBA) were active in the early swing phase (M4 in Fig. 4A), but in cases B, C and D were found to be associated to the RECT in the same synergy modules. The other muscles of the quadriceps (VASL-VASM) were included in different modules (Fig. 4: A M6; B M5; C M4), associated with hip extensors (GLUMAY-GLUM-TFL), all active in the late swing phase, between 80%–94% of the normalized cycle (Fig. 5B). Knee flexors were active in the swing phase, between 72%–92% of the normalized cycle (Fig. 5B).

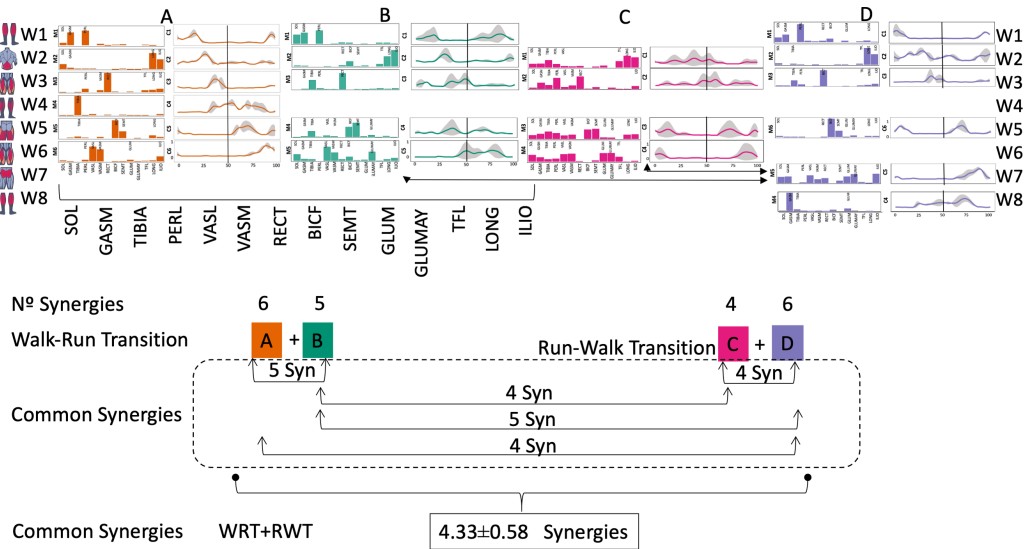

**Figure 4** Synergy similarities between transition gears. Cosim comparison > 0.6 assessed with test t or Wilcoxon ranking test. WRT and RWT shared on average 4.33 ± 0.58 synergies; 5 synergies were shared between A and B and between B and D; 4 synergies were shared between A and D, between C and D, and between B and C.Muscle Group illustrations source credit: https://stock.adobe.com/es/images/muscles-illustration-icon-set-it-included-the-workout-human-body-parts-anatomy-and-more-icons/600376660?asset_id=600376660. Standard license, creator: antto.

## DISCUSSION

Although W and R share the same synergy modules, differing only for the activation timing profile (*Cappellini et al., 2006*; *Hagio, Fukuda & Kouzaki, 2015*; *Ivanenko, Poppele & Lacquaniti, 2004*; *Pequera, Ramírez Paulino & Biancardi, 2021*), we demonstrated that the transition strides between the two gaits (WRT and RWT) are characterized by sudden reorganizations of both, modules and timing profiles.

Our sample (healthy, recreational male runners) did not differ significantly from those presented by *Cappellini et al. (2006)* and *Yokoyama et al. (2016)*, who likely included young, healthy adults as participants, given the basic nature of their locomotion studies. Like the participants in studies by *Pequera, Ramírez Paulino & Biancardi (2021)* and *Hagio, Fukuda & Kouzaki (2015)*, our sample probably included individuals capable of both walking and running, potentially encompassing a wider age range and physical condition. Perhaps the closest match in terms of prior experience is the study by *Santuz et al. (2017b)*, as their sample included individuals willing to run both shod and barefoot, suggesting a sample of recreational runners or individuals interested in running.

We will discuss in details the features, their implications and their possible explanations.

### Activation profiles

The analysis of EMG envelopes revealed different activation times during the transition strides, with respect to W and R. Indeed, transition strides are characterized by positive or negative accelerations and accommodations related to gait change.

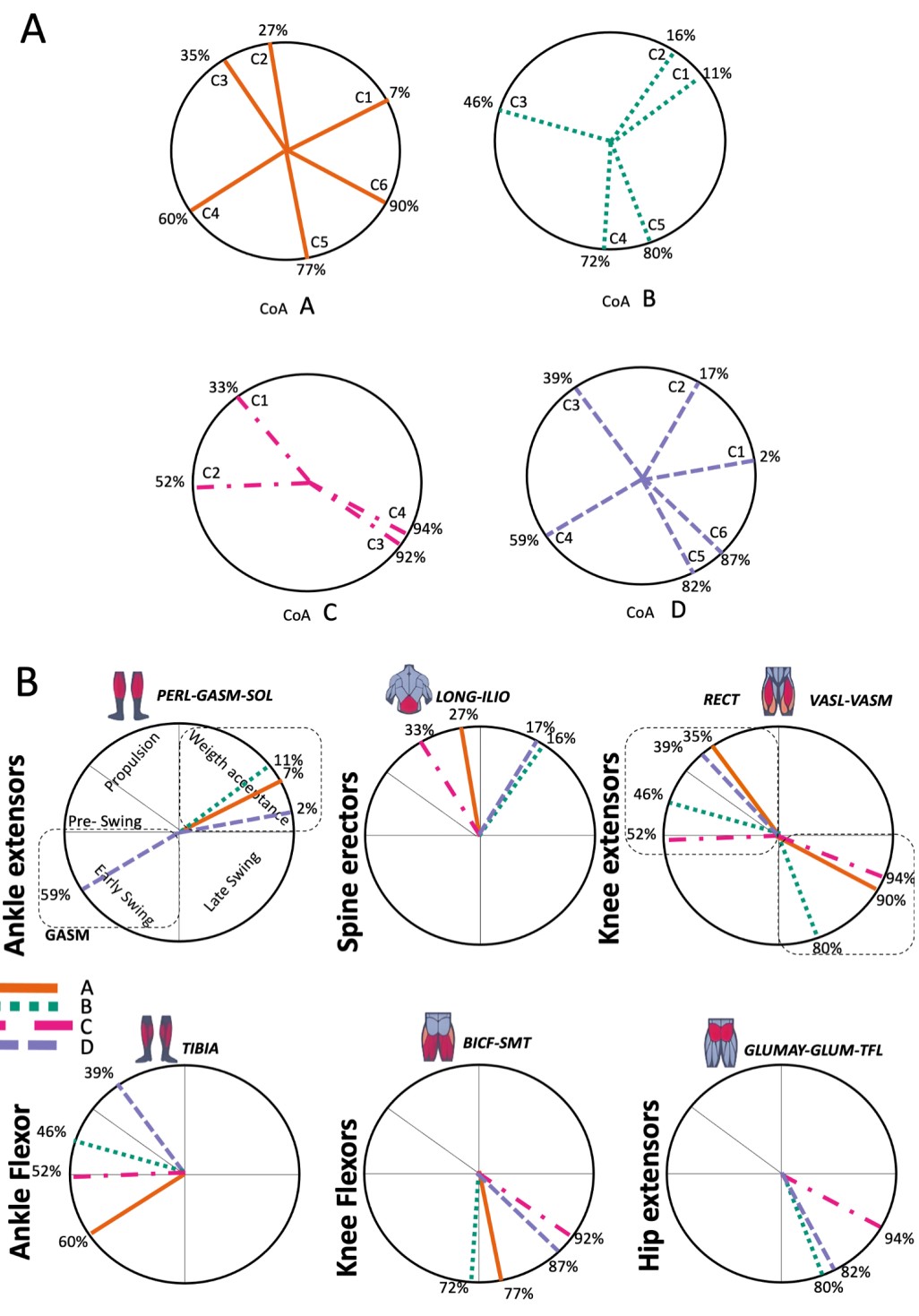

**Figure 5** **Activation center.** The gait cycle is represented by a unit circle, where the cycle begins at position 0 (0%) and proceed counterclockwise to the end at position 2pi (100%). 5A: CoA's of WRT (A, B) and RWT (C, D). 5B: CoA's according to the muscular groups. Colours represents the transition types. Muscle Group illustrations source credit: https://stock.adobe.com/es/images/muscles-illustration-icon-set-it-included-the-workout-human-body-parts-anatomy-and-more-icons/600376660?asset_id=600376660. Standard license, creator: antto.

ILIO is a trunk extensor muscle, and its early activation in WRT (A and B), compared to W and R (*Cappellini et al., 2006*), can be explained by the positive acceleration needed for this gait transition, which leads to adjustments in trunk position. In RWT the acceleration is negative, which would not need adjustments in the extension of the trunk.

Among hip extensors, GLUM and GLUMAY were activated similarly to W and R (*Cappellini et al., 2006*; *Pequera, Ramírez Paulino & Biancardi, 2021*; *Santuz et al., 2022*). Only in WRT (B) the GLUM began its activation in the stance phase, something that happens when pushing walking to its speed limits (*Cappellini et al., 2006*).

The knee extensors activation in RWT was similar to what occur during running (*Cappellini et al., 2006*). The late swing activation in transition A and throughout the swing in B in WRT was possibly a result of the rapid transition demand. On the other hand, it was observed in WRT and RWT that RECF activation occurred at the beginning of the cycle during slow walking, and it was towards the middle part during fast walking and running, placing itself close to the HS as a strategy for quick knee extension acceleration (*Cappellini et al., 2006*).

The knee flexor group displayed activations in RWT during stance and early swing, like running (*Cappellini et al., 2006*; *Pequera, Ramírez Paulino & Biancardi, 2021*; *Santuz et al., 2022*). This could be explained by the leg deceleration need before TO and flexion. However, in WRT, low muscular activity was found at beginning of A, like in knee extensors group. We also found that B brought the activation range closer, probably due to being a faster mode that mixed walking stance and running swing (Fig. 2).

A running like behaviour was found in tibialis anterior muscle during WRT and RWT (*Cappellini et al., 2006*). Activation increased with speed from the HS and before TO. Such high demand in transition would suggest a main role in ankle preparation and stabilization as increased muscular activation was detected in presence of knee functional instability (*Delahunt, Monaghan & Caulfield, 2007*).

The ankle extensors (SOL-GASM-PERL) had their maximum activation close to the 10%–25% of the walking cycle, differently from what exposed by other authors (*Cappellini et al., 2006*; *Pequera, Ramírez Paulino & Biancardi, 2021*). The maximum activation shifted to the beginning of the cycle while running. This may be explained by the effort required to take-off.

## Muscular synergies

It has been found using unilateral EMG that there are from four to six shared muscular synergies in locomotion for walking, running and skipping (*Cappellini et al., 2006*; *d'Avella, Saltiel & Bizzi, 2003*; *Pequera, Ramírez Paulino & Biancardi, 2021*; *Santuz et al., 2022*). Conversely, by using bilateral EMG eight to nine synergies were found for walking and running (*Hagio, Fukuda & Kouzaki, 2015*). In the study developed by *Hagio, Fukuda & Kouzaki (2015)*, they measured a RWT speed of $1.84 \pm 0.08$ m.s[1] (close to $1.98 \pm 0.70$ m.s[1] obtained in our work) and identified eight to nine synergies in spontaneous and forced transitions (Table 2). In our work using unilateral EMG including lower back muscles we obtained six and five synergies for A and B respectively in WRT. Similarly, four and six synergies were obtained for C and D in RWT (Fig. 4). We found that five synergies were

shared in WRT (A and B) and four in RWT (C and D). Our results were consistent with that of Hagio and collaborators (*Hagio, Fukuda & Kouzaki, 2015*), considering just one extremity.

The weight acceptance phase includes synergies with predominance of hip, knee, and ankle extensors, regardless locomotion type (*Cappellini et al., 2006*; *Mileti et al., 2020*; *Pequera, Ramírez Paulino & Biancardi, 2021*; *Santuz et al., 2017a*), including RWT (*Hagio, Fukuda & Kouzaki, 2015*) (Table 2). Our results, which included spinal erector muscles, were consistent with that scenario (Fig. 4, column C).

The propulsive phase is characterized by synergies where knee extensors lead (*Cappellini et al., 2006*; *Mileti et al., 2020*; *Pequera, Ramírez Paulino & Biancardi, 2021*; *Santuz et al., 2017a*) with the addition of knee flexors in WRT and RWT (*Hagio, Fukuda & Kouzaki, 2015*). Our results included spinal erectors in A,B, D and knee extensors in A. This could be explained by control strategy switching from walk to run and vice versa.

Synergies during the stance phase have been described slightly differently by different authors (Table 2): in W and R *Cappellini et al. (2006)* pointed to ankle flexors activation, while *Santuz et al. (2017a)* indicated column erectors, knee and ankle extensors, and *Yokoyama et al. (2016)* included hip, knee and ankle extensors; in WR transitions, flexors and extensors of knee on one side and ankle on the contralateral were suggested (*Hagio, Fukuda & Kouzaki, 2015*). Our results were consistent with some of the previous descriptions: knee and ankle extensors in WRT (A) (*Santuz et al., 2017a*); synergy on ankle flexors in running (B) (*Hagio, Fukuda & Kouzaki, 2015*). Different from previous work, in pre-swing (D) we found a new synergy including hip extensors and ankle flexors, which could be related to the sudden switch from run to walk in (D) transition.

A new synergy that includes knee and ankle flexors in both transitions was found in early swing (Fig. 5B). Previous works reported, in this position of the cycle, modules with ankle flexors (*Mileti et al., 2020*; *Pequera, Ramírez Paulino & Biancardi, 2021*; *Santuz et al., 2017a*), spinal erectors (*Cappellini et al., 2006*; *Yokoyama et al., 2016*) or knee extensors (*Pequera, Ramírez Paulino & Biancardi, 2021*). On the other hand, a synergy with spinal erectors and knee flexors was found in C, and ankle extensors in D (*Hagio, Fukuda & Kouzaki, 2015*).

To our knowledge, previous works reported synergies with only knee flexors during late swing in walking or running (Table 2) (*Cappellini et al., 2006*; *Mileti et al., 2020*; *Pequera, Ramírez Paulino & Biancardi, 2021*; *Yokoyama et al., 2016*). Ankle flexors synergies were additionally found when gait included skipping or transitions (WRT or RWT) (*Hagio, Fukuda & Kouzaki, 2015*; *Pequera, Ramírez Paulino & Biancardi, 2021*). Our WRT results showed a component that required knee extensors in A; ankle, knee and hip extensors in B and C; and spinal erector in D (Figs. 3 and 4).

Overall, transition types (A, D) revealed more synergy modules than types (B, C), where the two steps of the transition stride were more similar (both walking-like or both running-like). It would be reasonable to think that more synergy modules would be needed in transition strides including steps similar to both gaits. In those cases, the synergy modules would include fewer muscles, and their activation would be staggered over the stride time. Other studies displayed similar observations in transitions and acceleration (*Hagio, Fukuda*

**Table 2 Comparative table of synergies.** The table represents the synergy modules described by different authors per gait and gait cycle phase. Muscle Group illustrations source credit: https://stock.adobe.com/es/images/muscles-illustration-icon-set-it-included-the-workout-human-body-parts-a natomy-and-more-icons/600376660?asset_id=600376660. Standard license, creator: antto.

| Author | Gait | Stance | | | Swing | |
|---|---|---|---|---|---|---|
| | | Weigth acceptance | Propulsion | Pre-swing | Early swing | Late swing |
| *Cappellini et al. (2006)* | Walking and Running | M1  | M2  | M3  | M4  | M5  |
| *Santuz et al. (2017b)* | Shod and barefoot running | M1  | M2  | M3  | M4  | M5  |
| *Pequera, Ramírez Paulino & Biancardi (2021)* | Walking, Running, Skipping | M1  | M2  | | M3  | M4  |
| *Yokoyama et al. (2016)* | Walking and running | M1  M2  | M3  | M4  M7  | M5  | M6  |
| *Mileti et al. (2020)* | Walking | M1  | M2  | | M3  | M4  |
| *Hagio, Fukuda & Kouzaki (2015)* | Walking-to-Run and Run-to-Walking transition | M1  M4  | M9  | M8  M3  | M7  | M2  M5  M6  |

 Spine Erectors  Hip Extensors  Knee extensors  Knee flexors  Ankle Extensors  Ankle Flexor

& *Kouzaki, 2015*; *Kibushi et al., 2018*). Moreover, more synergy modules were detected in WRT compared to RWT. In the latter transition type less synergies were required, but with a larger number of active muscles in each module (fig 3 y 4). It could be possible that CNS would make use of less synergies and a large number of co-activated muscles (agonists and antagonists) to have a larger number of muscles involved in a challenging motor activity, and to improve the intermuscular efficiency. Larger muscular co-activation would be needed to absorb negative acceleration and reduced oscillations of the centre of mass.

It has been demonstrated that, regardless of speed, walking, running and skipping share the same synergies (*Cappellini et al., 2006*; *Pequera, Ramírez Paulino & Biancardi, 2021*). This may suggest that transition strides could be an exception with different regulation at CNS level, and "special" kinds of synergies.

## Synergy temporal patterns

During the stance phase, the transition modes A, B and D shared the temporal sequence of activation of the first three synergy modules: M1/C1 (ankle extensors), M2/C2 (spinal erectors) and M3/C3 (knee extensors) (Fig. 4). The position of the CoA showed modifications according to the transition types (Fig. 5A), in agreement with the differences observed in W and R (*Cappellini et al., 2006*; *Pequera, Ramírez Paulino & Biancardi, 2021*). During the swing phase, the temporal sequence of WRT (A and B) were more similar to each other than to the RWT transition types (Fig. 4). The behaviour of synergies during swing phase can characterize the transition between W and R and vice-versa (*Kibushi et al., 2018*). Distinct from the others, in RWT C were observed synergies starting in the late swing phase and extending across the stance phase (Figs. 4 and 5A). The reduction of the number of synergies, together with an extension of the temporal activation pattern has been observed in situation of reduced stability (*Clark et al., 2010*). Therefore, RWT in particular can be challenging for gait stability.

Looking to the muscular groups, they displayed consistent sequences of CoA's regardless of the transition type. The cycle was initiated by the knee extensors, then the spinal erectors, knee extensors and ankle flexors, and finally the ankle flexors and hip extensors (Fig. 5B).

The section of temporal pattern across the ending and beginning of the gait cycle (from 80% to 15–20%) in walking, running and skipping was occupied by synergy modules including knee and hip extensors (*Pequera, Ramírez Paulino & Biancardi, 2021*). These muscular groups (*Vastus medialis* and *lateralis*, *Gluteus maximus* and *medius*) participated in the transitions A, B and C, with the CoA at 80% to 90% of the cycle (Fig. 5B). We observed that this behaviour was consistent with that of the trailing limb of skipping (*Pequera, Ramírez Paulino & Biancardi, 2021*).

Ankle extensor group activation, with CoA at 10% to 20% in WRT and RWT, was consistent with running (*Santuz et al., 2017a*; *Yokoyama et al., 2016*) and skipping (*Pequera, Ramírez Paulino & Biancardi, 2021*) but distinct from walking (*Cappellini et al., 2006*; *Mileti et al., 2020*; *Yokoyama et al., 2016*). A distinctive finding that links running, skipping and transitions would be the earlier CoA of this muscle group in relation to walking. Ankle extensor group (*Tibialis anterior*) and knee extensor (*Rectus femoris*) CoA's were positioned

at 60% to 85% of the gait cycle in walking and running (*Hagio, Fukuda & Kouzaki, 2015*; *Santuz et al., 2017a*), and at 30% to 50% of the gait cycle in skipping (*Pequera, Ramírez Paulino & Biancardi, 2021*). In our results both muscular groups revealed a behaviour similar to skipping in both WRT and RWT, apart from condition A in WRT, more similar to running. This general behaviour could explain the muscular preparation of the distal segments for the transition from walking to running, and partially the complexity in the rehabilitation processes of these muscles in pathologies of the central nervous system (*Sánchez Milá et al., 2022*).

The CoA of knee flexors was located at 90% of the gait cycle in walking and running (*Cappellini et al., 2006*; *Mileti et al., 2020*; *Santuz et al., 2017a*; *Yokoyama et al., 2016*), and at 75% to 90% of the gait cycle in skipping (*Pequera, Ramírez Paulino & Biancardi, 2021*). Again, the CoA of WRT was more similar to skipping (72% to 77%) than to walking and running. On the other hand, the CoA of RWT was between walking/running and skipping (87 to 92%).

The CoA of spinal erectors was located at 50% of the gait cycle in walking and running, in order to control trunk acceleration (*Cappellini et al., 2006*; *Kibushi et al., 2018*). We found an earlier activation of this muscular group in WRT and RWT (16% to 33%). This finding confirms a preparatory phase for transition control and execution.

These particularities reinforce the idea that transitions are specific and special events for neuromuscular control, that have traces of symmetric and asymmetric control according to the stress level imposed. These results agree with previous investigations that proposed that the location of the CoA would be an indicator of the intense activation of muscle synergies during speed regulations (*Kibushi et al., 2018*).

## Practical applications

The functionality of the knee extensor and ankle flexor muscular groups are some of the main problems of patients affected by neurological lesions of motor neurons (*Diner et al., 2023*; *Sánchez Milá et al., 2022*). We have identified the main role of *Tibialis anterior* muscle in three of the four transition modes shown. Furthermore, there are synergy modules in WRT or RWT that responds almost exclusively to it in stance to swing. We also identified synergy modules that included iliocostal and paraspinal muscle activations in the four transition modes which main function was for support. These findings could be interesting for recovering of pathological conditions such as hernias of the nucleus pulposus or cerebrovascular accidents, but also in athletes with high transition traffic such as tennis players, basketball players and trail runners.

We believe that the main contributions of this work were: (1) to identify that A and D modes involved greater neuromuscular specificity to activate isolated muscles; and (2) that B and C demands a co-activation of a larger number of muscles. In this context, it seems necessary to recommend the use of transitions with accelerations and decelerations including changes with other modes of locomotion (running, walking or skipping). This could impose increased neuromuscular stress on the CNS, in different clinical or sports contexts.

## Limitations of the study

Our research highlighted some limitations that should be addressed in future investigations. The calculation of the step kinematics has been carried out stride by stride, under the assumption that the vertical position of the heel marker would indicate the beginning of a stride (heel strike). This method has been validated with a comparative methodology including walking, running and skipping at different speeds, with a maximum error of less than 2% (*Cappellini et al., 2006*). It is known that progressive increases in running speed can change foot strike pattern, moving from rearfoot to midfoot and forefoot strikes (*Cheung et al., 2017*), which could result with an incorrect detection of the beginning of a stride. However, only 5% of people running with shoes on a treadmill in a wide range of speed use midfoot strike, none (0%) use forefoot strike (*Lai et al., 2020*). Further, it is worth considering that we are working with near-transition speeds, that is low running speeds. Finally, we recognize the foot strike detection as a point of possible improvements in future research including variable gaits and variable speed tests.

Even if W and R are symmetrical gaits, bilateral EMG analysis could be included to more comprehensively capture muscle synergies and their organization during locomotion transitions. Our research was carried out with trained men. It should be recommendable to design different investigations including different kind of participants. This would improve the generalizability of the results. It would be also interesting to examine transitions under a variety of experimental conditions, including different speeds and terrain types, to assess how these factors influence muscle synergies. Conducting longitudinal studies would allow for the observation of how muscle synergies change over time, especially in populations with neurological conditions or high-performance athletes. Finally, comparing walking-to-running and running-to-walking transitions with other types of locomotion, such as jumping or stair climbing, would help to better understand the specificities of the analyzed transitions.

## CONCLUSIONS

Characteristics of transitions and synergies

- Walk-to-run transitions (WRT) and run-to-walk transitions (RWT) involve significant reorganizations of muscle activation patterns.
- There are differences in muscle activation patterns among the different types of transitions (A, B, C, D).
- Four to six synergy modules are involved during transitions, compared to the four to five involved in W and R.
- Transitions involving more dissimilar steps (A and D) require a greater number of muscle synergies.
- Muscle activation patterns within synergies show specific variations during different phases of the transitions.
- The temporal distribution of synergies (centers of activation) reveals differences between the different types of transitions.

Implications

- The results suggest that walk-to-run transitions represent a particular challenge for the central nervous system, requiring rapid and precise adjustments in muscle coordination.
- Understanding the muscle synergies involved in these transitions can contribute to the development of more effective therapeutic and sports training interventions.
- Further studies are needed to investigate in depth the differences in muscle synergies between individuals and in different environmental conditions.

In conclusion, the study provides evidence of the complexity of walk-to-run and run-to-walk transitions in terms of muscle synergy reorganization. The results obtained can contribute to a better understanding of the neural mechanisms underlying these transitions and can potentially be applied to various fields.

## ACKNOWLEDGEMENTS

The authors acknowledge Mateo Rodríguez and Gonzalo Giannecchini for its support during data collection. The authors thank the Running Clubs of Paysandú-Uruguay.

### Funding

Samuel Vergara received support from Proyecto Ingenieria 2030 (ING222010004) for the APC. The funders had no role in study design, data collection and analysis, decision to publish, or preparation of the manuscript.

### Grant Disclosures

The following grant information was disclosed by the authors:
Proyecto Ingenieria 2030: ING222010004.

### Competing Interests

The authors declare there are no competing interests.

### Author Contributions

- Leonardo Lagos-Hausheer conceived and designed the experiments, performed the experiments, analyzed the data, prepared figures and/or tables, authored or reviewed drafts of the article, and approved the final draft.
- Samuel Vergara analyzed the data, authored or reviewed drafts of the article, and approved the final draft.
- Victor Munoz-Martel analyzed the data, authored or reviewed drafts of the article, and approved the final draft.
- Germán Pequera performed the experiments, analyzed the data, authored or reviewed drafts of the article, and approved the final draft.
- Renata L. Bona conceived and designed the experiments, performed the experiments, analyzed the data, authored or reviewed drafts of the article, and approved the final draft.

- Carlo M. Biancardi conceived and designed the experiments, performed the experiments, analyzed the data, prepared figures and/or tables, authored or reviewed drafts of the article, and approved the final draft.

### Human Ethics

The following information was supplied relating to ethical approvals (i.e., approving body and any reference numbers):

Ethics Committee of the CENUR Litoral Norte—Universidad de la República (Exp. # 311170-000921-19).

### Data Availability

The data and R script are available in the Supplemental Files.

### Supplemental Information

Supplemental information for this article can be found online at http://dx.doi.org/10.7717/peerj.18162#supplemental-information.

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
