# Peer review of "Muscle synergies during the walk-run and run-walk transitions"

_PeerJ, doi:10.7717/peerj.18162_

## Round 0.1 · original submission · Major Revisions

Dear Authors:

Thank you for submitting your manuscript to PeerJ journals. After a high-quality peer review, we consider "major reviews". Please read and respond to the reviewers.

Best regards

Dr. Manuel Jiménez

·

Basic reporting

1) This study aimed to examine the behavior of muscle synergies in different stance and swing variations of walking-running (WRT) and running-walking (RWT) transitions. The objective would benefit from clearer wording, as "study" reads too broadly. In fact, the authors seem to investigate whether the transition from walking to running can elicit different muscle synergies.

2) The introduction section reviews previous studies on locomotion synergies, but it takes five paragraphs to mention the main problem of the study. When the authors state that there are still open questions regarding muscle activations and synergies, especially when different modalities of transition are involved, there is a lack of clear application of the results. I suggest that the authors contextualize their potential results considering previous studies on this topic. How could their findings change or add to future research with similar goals?

Experimental design

3) The overall protocol seems to be well conducted, and the data validity can be ensured. I do not consider the opening sentence stating where the study was developed to be relevant. This information can be noted from the authors' affiliation, and in terms of scientific methods, where the study was developed should not be a major methodological issue worthy of reporting.

4) The authors describe the inclusion of 24 participants, but I did not find mention of sample size criteria. I understand the challenges of developing research with this design, but to ensure the proper validity of the findings, a statistical rationale for the sample size is needed.

5) Foot strike is a major concern in this study as it may influence the outcomes of EMG assessments. The authors refer to a previous publication to explain how foot strikes were determined. The referenced paper states that "analyses were performed stride by stride, and a threshold on the vertical position of the left heel marker, with respect to the belt, was used to detect heel strike." Such an approach, relying on the kinematics of one marker, may lead to bias depending on the landing pattern, which can significantly change due to changes in gait speed. How did the authors ensure that a heel strike pattern was consistently found among the participants and conditions?

6) Regarding foot strike identification, the referenced paper reports this method for the left leg, while in the study, EMG was recorded from the right lower limb. Was foot strike defined the same way for both sides?

Validity of the findings

7) The authors dedicated part of the discussion to changes in activation times, but such results are somewhat expected due to the experimental design. Was there significant intersubject variability in these outcomes that would be worth noting in the discussion? Do any activation time patterns relate to the synergies found?

8) In general, I noted that the discussion contains much description of previous research but lacks in comparing and contrasting findings. Aspects that the authors may consider to improve the discussion include the nature of the samples, as not all studies consider participants with similar characteristics. Additionally, previous experiences with running and physical fitness should be noted in the discussion.

9) I would suggest the authors make the conclusions more straightforward.

10) Please consider acknowledging some limitations noted during the course of the study's development.

Reviewer 2 ·

Basic reporting

The authors aim to identify muscle synergies during walking-to-running transitions, expecting to discover distinct recruitment patterns or different numbers of synergies. However, their findings reveal no significant difference between simple walking/running and walking-to-running transitions.

The article currently presents a descriptive list of muscle patterns for walking, running, and their transitions, lacking a theoretical framework or rigorous grounding in prior research. The absence of dependency in muscle synergies during transitions could be intriguing, but the analytical methods used to identify these synergies are flawed. For example, normalizing each stride separately for the stance and swing phases distorts the variability, which might influence the interplay of various muscle activations. This method erroneously assumes independence between consecutive strides, contradicting recent findings by Mangalam et al. (2024) in PLoS One.

Specific details on synergy analysis, such as the composition of matrices, are missing. Additionally, lines 177-179 are unclear and need revision.

I recommend that the authors share their data and code as supplementary material or host it at a permanent link to enable replication.

The authors should provide clear hypotheses and predictions regarding muscle synergies in this context. They should explain why changes in synergies are expected during walking-to-running transitions and specify the nature of these changes, including total numbers and the number of muscles involved.

My primary concern is that the manuscript currently reads more like a data analysis exercise rather than an exploration of the mechanics of walking-to-running transitions. The authors need to connect their expectations and results to existing knowledge on the physiology and biomechanics of walking and running.

Additionally, several sentences are awkward or incomplete and require revision. The manuscript would benefit from more detailed figure captions, particularly for the final figure, to facilitate reader comprehension.

Experimental design

.

Validity of the findings

.

---

## Round 0.2 · Minor Revisions

Dear Authors:

The manuscript has improved substantially, so we thank you for your patience and your interest in improving it. Still, some minor revisions are necessary. Please take note of the comments of reviewer #1

Regards

Dr. Manuel Jiménez

·

Basic reporting

no comment

Experimental design

The authors have responded to my questions and overall I am satisfied. I still would have some concern regarding the foot strikes event identification by the reasons I mentioned in my first review. If they could provide further information about the possible changes in foot landing pattern due to changes in speed, it would be good.

Validity of the findings

Of course there are some issues in the study of muscle synergies as also pointed out by the other reviewer with which I agree in parts. However, I consider that this study has a clear contribution and may help other researchers addressing this topic to further advance in the field.

Additional comments

no comment

Reviewer 2 ·

Basic reporting

No comment

Experimental design

No comment

Validity of the findings

No comment

Additional comments

No comment

---

## Round 0.3 · accepted · Accept

Dear Authors:

your manuscript "Muscle Synergies during the walk-run and run-walk transitions" has been accepted for publication in PeerJ.

Congratulations

Dr. Manuel Jiménez

·

Basic reporting

I have no further comments

Experimental design

I have no further comments

Validity of the findings

I have no further comments

Additional comments

I have no further comments